# OpenReview forum: "Cross-Architecture Model Diffing with Crosscoders: Unsupervised Discovery of Differences Between LLMs"
_ICLR.cc/2026/Conference — Submitted to ICLR 2026_

### Official Review · Reviewer_DCdW · 2025-10-24

**Soundness:** 2
**Presentation:** 2
**Contribution:** 3
**Rating:** 4
**Confidence:** 4

**Summary:**

This paper presents the first application of crosscoders for model diffing between different model architectures, highlighting interesting ideological features which differ between models in different families. The authors introduce Dedicated Feature Crosscoders, which partition the feature dictionary to enforce model-exclusive features rather than identifying these post-hoc based on feature decoder norms. They find that DFCs are more effective at identifying model-exclusive features in both a toy model, where the ground truth features are known, and in real LLMs. The exclusivity of features is validated using model-stitching to assess feature transferability between models.

The application of crosscoders between architectures is a useful demonstration of their potential for auditing behavioural differences between models, and this work presents some helpful methods, such as the window expansion algorithm for activation alignment, to enable this. The DFC method also enables improved identification of model-exclusive features, which is quite well validated with the toy and real model setups and has applications for both cross-architecture and single-architecture crosscoders. This is generally an interesting and novel piece of work. However, the analysis is weak in several places, which causes me to weakly recommend rejection. Particularly, the 'exclusivity score' which underlies most of the results should be better validated, and the feature analysis should be more extensive beyond the cherry-picked ideological feature examples.

**Strengths:**

This work presents a useful case study of a novel application of crosscoders (cross-architecture model diffing). It also presents a simple and novel modification to improve their performance in identifying model-exclusive features more broadly.

The use of the toy model with a ground truth is a compelling validation of the DFCs improved ability to extract model-exclusive features. Including results for 2 different model pairs improves validity, though more models across different sizes would be valuable (including potentially diffing models of different sizes in the same family).

The paper is generally well presented and clear, though several key aspects of the method are only included in the Appendices (see weaknesses).

**Weaknesses:**

The exclusivity score is an important part of the validation for DFCs, and some more details of how this is calculated should be given in the main text. Similarly, for the toy model, I would recommend including some additional information regarding how the uniform sampling of the ground truth concepts in the main text to improve reader understanding.

The exclusivity score seems poorly validated. Did you manually validate a good number of the LLM judged examples, and can you include some examples of steered behaviours and their similarity scores in the Appendices? Also, how did you select the maximum steering magnitudes and how did you ensure that the range tested was sufficient to elicit behavioural changes? (For example, you could have steered to a magnitude where 80% of responses were still coherent, and compared behaviours in these responses, rather than instructing the judge to ignore coherence degradation).

The false positive rate ‘recovering shared concepts’ seems extremely high (Figure 3, right). When applying DFCs for behavioural diffing in safety auditing, identifying shared features as model-exclusive does not necessarily seem to be a favourable trade-off. It would be useful to have some discussion of how this trade-off arises and whether the issue could be mitigated.

The analysis focuses heavily on the ideological model-exclusive features, which are clearly heavily cherry-picked. I appreciate the analysis of LLM identified important features in Appendices C.5 and C.6, but this seems weak given the claims that this technique can be generally useful for model-auditing.

Minor comment - check all of your quotation mark formatting (e.g. line 40)!

**Questions:**

Is there a reason why the DSF crosscoder scores are not included in Figure 4? This would be useful to see.

In Section 2.2 you describe allocating 1-5% of total features to the model-exclusive partitions, but seem to mainly include results for the 5% DFCs with only minimal results from 1% DFCs in A C.4.  Can you include some more details of how this affected results, e.g. with 1% DFC versions of Figure 4? If the 5% version gave the best results, did you try higher allocations?
Regarding the analysis of model-exclusive features - what prompt was used to ask Claude to flag important features? (How was ‘relevance’ defined?)

You say that the  “vast majority of features flagged as important relate to general concerns” and may be false positives. This seems to undermine any reliable auditing application. Have you tested any methods for identifying these (the approaches in Section 3.2.1 don't seem to address the 'recovering shared concepts' issue)? Otherwise could you include a more detailed discussion of how this method could still be useful when the features are unreliable?

To avoid focusing on cherry-picked results, can you include a selection of feature explanations for randomly selected model-exclusive features for each model, along with their exclusivity scores?

See also the questions in weaknesses.

---

> ### Author Response · Authors · 2025-11-25
>
> We thank the reviewer for their constructive feedback and for recognizing the novelty and utility of applying crosscoders to cross-architecture model diffing. We appreciate your specific suggestions regarding validation and robustness, which have significantly strengthened our manuscript.
>
> 1. Validation of the Exclusivity Score
> You raised important questions regarding the validation of our primary metric. We have addressed these in the updated paper:
>
> - Manual Validation: We performed manual validation of the LLM judge's ratings. As noted in Section 4.2.2, we validated 25 ratings and found strong alignment with human judgment. We have added Appendix B.4.7 (Steering-Based Exclusivity Score: Representative Examples), which shows examples across the scoring spectrum (1-5) alongside the LLM judge's reasoning, demonstrating that the score accurately captures behavioral divergence.
>
> - Steering Magnitude: We clarified in Appendix B.4.3 that we scaled steering vectors based on the maximum activation (s_max) observed in the dataset. We then manually selected steering strengths up to 10, a range where we observed all features eventually exhibiting coherence degradation. While we acknowledge this ad hoc selection is not the most principled approach, we believe it effectively demonstrates the feature's causal effect on behavior across a sufficient range of intensities.
>
> - Methodological Details: We have moved the core definition of the exclusivity score to the main text (Section 4.2.2) and provided the full algorithmic details in Appendix B.4.6.
>
> 2. False Positives and Cherry Picking
> You rightly pointed out the high false positive rate (Figure 3) and concerns about cherry-picked ideological features. We have updated Section 4.3 to explicitly address these trade-offs by framing the method as a high-recall pre-screening tool.
>
> We added the following text to the paper to transparently define this workflow:
>
> "We note that **most** features identified as model-exclusive do not capture meaningful behavioral differences (see Appendix C.5). Consequently, we propose crosscoder model diffing primarily as a high-recall pre-screening tool designed to surface potential areas of divergence for further investigation. The specific features presented below were identified through a multi-step auditing pipeline: first, we filtered for potentially concerning features based on automated explanations (see Appendix B.4.2); next, we validated their causal effect and behavioral relevance using activation steering and external evaluations. We recommend this "screen-and-verify" workflow for future safety auditing. Notably, while the ideological alignment features discussed below were hypothesized based on prior work (Rager et al., 2025), the copyright refusal feature in GPT-OSS-20B represents a meaningful behavioral difference that we were unaware of a priori. This discovery highlights the potential of crosscoder model diffing to uncover unknown unknowns in model behavior."
>
> To support this, we emphasize two new additions:
>
> - Random Feature Sampling: To address cherry-picking, we added Appendix C.5, which analyzes randomly sampled exclusive features. This confirms the DFC captures valid, non-ideological structural differences (e.g., specific error handling patterns).
>
> - Discovery of "Unknown Unknowns": We highlight that while our ideological features were hypothesized, the Copyright Refusal feature in GPT-OSS-20B represents a meaningful difference we were unaware of a priori. This discovery validates the potential of this workflow to uncover unknown unknowns despite the noise.
>
> 3. Robustness and Hyperparameters
> - DSF in Figure 4: We omitted the Designated Shared Feature (DSF) crosscoder from Figure 4 (Real Models) due to the high compute cost (requires training separate stitching layers) and its poor performance in the Toy setting (Figure 3).
> - Partition Sizes: We added a detailed robustness analysis in Appendix C.3 and Table 7. We found that while 1% partitions are sufficient for broad features (e.g., "CCP Alignment"), larger partitions (5%) were necessary to reliably recover granular narratives.
> - Feature Importance Prompt: We have included the full prompt used to flag important features in Appendix B.4.2.
>
> We also thank you for the suggestions regarding these points
> - Toy Model Details: We have added details about the uniform sampling of ground truth concepts to Section 3.3.
> - Formatting: We have fixed the quotation mark formatting throughout the paper.
>
> We hope these revisions/explanations address your concerns and that you will consider adjusting your score accordingly.
>
> We remain available to answer any additional concerns/comments that you might have.

---

> > ### Comment · Reviewer_DCdW · 2025-11-25
> >
> > Thank you for the response.
> >
> > I especially appreciate the honesty with which you now define the limitations of the methods. However, I still see the extremely high false positive rate as a major limitation of the method, which greatly reduces its utility for safety auditing. I would encourage the authors to revisit the DFC technique in order to reduce this.

---

> > > ### Author Response · Authors · 2025-11-26
> > >
> > > We agree that the high false positive rate is a significant limitation and a clear area for future improvement. However, we respectfully disagree that the method is not currently useful for identifying "unknown unknown" features.
> > >
> > > As proof of utility, we point to the fact that the method successfully identified CCP alignment and Copyright Refusal in a completely unsupervised manner. These are concrete, meaningful behavioral differences that were successfully recovered despite the noise. We believe the successful recovery of these features demonstrates that the method provides high utility as a high-recall, unsupervised discovery tool. Does the reviewer agree that this capability offers value for safety auditing contexts?

---

> > > > ### Comment · Reviewer_DCdW · 2025-11-27
> > > >
> > > > While I agree that these features are interesting, I do not find them to offer a notable safety-enhancing value, particularly given the overhead of training crosscoders. While cross-architecture diffing offers some value from a safety auditing perspective, I find this to be low relative to diffing across finetunes of a model, or studying a single model. It is quite plausible to me that an SAE paired with automated interpretability and flagging of features could have detected these features.

---

### Official Review · Reviewer_4fUV · 2025-10-27

**Soundness:** 1
**Presentation:** 1
**Contribution:** 1
**Rating:** 2
**Confidence:** 3

**Summary:**

This paper presents a technique for model diffing using an architecture (named DFC) focusing model-exclusive concepts. The evaluation results indicate that the proposed technique may help identify model-exclusive concepts in terms of dictionary sizes. Specifically, this paper also provides empirical results of diffing two pairs of models.

**Strengths:**

This paper presents an alternative architecture for model diiffing.

**Weaknesses:**

1. The study reported in this paper is not well-motivated. It is unclear why we need to diff models of different architectures. Diffing base models with their fine-tunes may help us understand the impacts of the fine-tuning. But this need is obviously not applicable for cross-model diffing.
2. The main part of this paper focuses on identifying model-exclusive features. First, I think identifying model-exclusive features is not specific to cross-model diffing, but also applicable to diffing base models and their fine-tunes. Second, whether the diffing should identify model-exclusive features or shared features may depend on the purpose of diffing. That is to say, we may not claim that identifying model-exclusive features is superior to identifying shared features.
3. The technical contribution is unclear. The DFC architecture is proposed in Section 2.2 (page 3) with its illustration (i.e., Figure 2) on the top of the same page. This part is too concise for readers to understand the real differences of the DFc architecture.
4. The evaluation is unsatisfactory. In my opinion, to demonstrate the superiority of DFC, we need to show that DFC achieves better results than standard crosscoders when performance same tasks. The reported evaluation focuses on only model-exclusive features, making the whole evaluation unconvincing.

**Questions:**

1. Why do we need to diff models of different architectures?
2. Why are model-exclusive features more important than shared features?
3. Is DFC more useful than standard crosscoders in any real tasks?

---

> ### Author Response · Authors · 2025-11-24
>
> We thank the reviewer for their excellent suggestions. Your comments highlighted a crucial need to better motivate the specific utility of cross-architecture diffing, which we have now addressed in the revised paper.
>
> 1. Motivation for Cross-Architecture Diffing
> You asked why we need to diff different architectures rather than just comparing base models to their fine-tunes. The main reason is that new releases are often not just finetuned versions of existing models: In the current AI landscape, major model updates (e.g., GPT-3 to GPT-4, or Llama 2 to Llama 3) almost always involve architectural changes (different layer counts, dimensions, or attention mechanisms). To understand what changed behaviorally between generations (from a safety auditing perspective), we fundamentally need methods that work across these boundaries.
>
> While we agree that comparing a base model to its fine-tune is valuable (and have noted this in our Related Work/Limitations), the focus of our work is on the cross-architecture setting
>
> 2. Value of Model-Exclusive Features vs. Shared Features
> We appreciate this thoughtful perspective. We do not claim that model-exclusive features are intrinsically "superior" to shared features for all purposes. However, we argue they are uniquely critical for the specific task of safety auditing, for two key reasons:
> - The "Version Control" Analogy: As mentioned in our introduction, we view this method as analogous to version control in software engineering. When a developer reviews a pull request, they focus on the diff (what changed/exclusive features) rather than the entire codebase (what is shared). Similarly, auditors need to identify emergent capabilities or regressions that were not present in the previous version.
> - Efficient Auditing: Shared features, by definition, represent behaviors common to both models. If a behavior is shared, prior evaluations likely cover it. Exclusive features represent the "difference": the new behaviors where fresh risks (like the emergent copyright refusal we found in GPT-OSS-20B) are most likely to hide. Focusing on exclusive features allows auditors to prioritize their limited attention on areas of highest uncertainty.
>
> 3. Comparison against Standard Crosscoders on Real Tasks
> You asked if DFC is more useful than standard crosscoders on real tasks. We argue that discovery of specific behavioral differences is the primary "real task" here (from a safety auditing perspective), and we provide evidence that DFC performs better at it:
>
> - Quantitative (Recall): Figure 3 shows the DFC significantly outperforms standard crosscoders in concept recovery recall on the toy model. Figure 4 shows DFC features achieve significantly higher exclusivity scores on real models.
> - Qualitative (Real-World Discovery): Most importantly, we ran additional seeds and partition sizes for the DFC on Llama vs. Qwen (added in Table 7). This analysis shows that the standard crosscoder failed to find granular features (e.g., specific "Taiwan status" or "Hong Kong politics" narratives) that the DFC successfully identified across multiple runs. This superior sensitivity to fine-grained behavioral differences demonstrates the DFC's practical utility over the standard architecture.
>
> 4. Technical Clarity
> We apologize for the brevity in the original text. We have significantly expanded Section 3.2 to explicitly detail the specific objective function of the DFC (Equation 2), ensuring the technical contribution is rigorous and clear.

---

> > ### Comment · Reviewer_4fUV · 2025-11-25
> > **Response to the authors' rebuttal**
> >
> > Dear Authors,
> >
> > Thank you for the rebuttal. After reading your rebuttal, I have an impression that you and I are talking about different things. What you are talking is "We aim to do something different and we show evidence that we have done it." What I am asking is "Please convince me that the something different is really important and beneficial." You may provide either strong arguments or empirical results as the answers to my questions. In your paper and rebuttal, the different something is crosscoder diffing and identifying model-exclusive features. But I cannot find satisfactory answers in either the paper or the rebuttal.
> >
> > Yours,
> >
> > Reviewer 4fUV

---

> > > ### Author Response · Authors · 2025-11-25
> > >
> > > Thank you for your feedback. We appreciate your request to clarify why this method is important and beneficial.
> > >
> > > The core benefit of cross-architecture model diffing is described in the introduction: its ability to find "unknown unknowns", i.e., behaviors we didn't know we needed to test for.
> > >
> > > 1. Why we need Crosscoder Model Diffing
> > > Standard evaluations (benchmarks, red-teaming) require us to know what to look for ahead of time. Model diffing is unsupervised: it highlights differences between models automatically. This acts as a high-recall "prescreening" tool that points safety researchers toward potential risks they might otherwise miss.
> > >
> > > 2. Why it must be Cross-Architecture
> > > Until now, model diffing was limited to comparing base models to their finetunes. However, the most important comparisons in AI safety are often between completely different models (e.g., comparing a new open-weights model to a proprietary safety-tuned one). Since new releases are often novel architectures, cross-architecture methods are essential to make model diffing widely applicable.
> > >
> > > 3. Empirical Examples of Benefit
> > > Our paper demonstrates this utility by identifying specific features such as Copyright Refusal in GPT-OSS-20B and CCP Alignment in Qwen3/Deepseek-R1 that correspond to concrete and very specific differences in model behavior. Crucially, these differences might have been missed by external evaluations which were not explicitly looking for them. This demonstrates the potential of our method as an unsupervised discovery tool.
> > >
> > > We believe this ability to audit models for unknown behaviors across different architectures fills a gap in the current safety toolbox.
> > >
> > > Could the reviewer kindly point us towards what part of this they disagree with?

---

### Official Review · Reviewer_H3Af · 2025-10-31

**Soundness:** 4
**Presentation:** 3
**Contribution:** 4
**Rating:** 8
**Confidence:** 2

**Summary:**

The authors apply crosscoders to cross-architecture model diffing and develop a modification of crosscoders (Dedicated Feature Crosscoders) that help find features that are exclusive to one model as opposed shared features.

**Strengths:**

- Clear motivation: model diffing to discover unknown behaviors not covered by evaluation suites
- Novel approach, very timely and original
- Clear explanations of shortcomings of previous methods

**Weaknesses:**

- Section 3.2.1. includes too much detail
- The motivation for the design change of DFC's is not clear enough
- A feature is not the same as a (propensity for a certain) behavior. How do you make sure that there are no other semantically different features that the method does not catch - features and concepts are not a 1:1 match
- Section 3.3 is the most interesting but also entirely qualitative; are there any quantitative metrics you might report as well?

**Questions:**

- Why is the false-positive rate expected to be increased? Is this generalizable?

---

> ### Author Response · Authors · 2025-11-24
>
> We thank the reviewer for their support and helpful suggestions regarding the manuscript's presentation and clarity.
>
> 1. Section 3.2.1 Detail
> Thank you for the excellent suggestion! We agree that the toy model section was overly detailed. We have streamlined the text in Section 4.2.1 and moved the implementation details to Appendix B.2, which improves the flow of the paper.
>
> 2. Design Motivation
> Thanks for the comment. We realize that the motivation wasn't described well enough and have clarified the motivation in Section 3.2. Specifically, we explicitly describe the "competition for capacity" in standard crosscoders: because shared features reduce reconstruction error for two models, they are favored by gradient descent over exclusive features. The DFC’s partitioning structurally protects model-exclusive features from this competition.
>
> 3. Features vs. Behaviors
> This is an important distinction. We have updated Section 4.3 to be transparent about this limitation. We explicitly state that most of the model-exclusive features identified by the DFC do not correspond to meaningful differences in model behavior. Consequently, we propose crosscoder model diffing primarily as a high-recall prescreening tool designed to surface potential areas of divergence. We describe our "screen-and-verify" workflow: using the DFC to cast a wide net, then validating causal effect and behavioral relevance using activation steering to filter out the majority of non-meaningful features.
>
> 4. Quantitative Metrics for Section 3.3.
> Though section 3.3 is mostly qualitative, the quantitative metric comes from the number of features corresponding to meaningful differences in behavior found by the DFC as opposed to the standard crosscoder. We have run additional seeds and additional partition sizes to demonstrate the robustness of our approach as well as a comparative table (Table 7) showing the features found by the DFC with different partition sizes vs the standard crosscoder. The conclusion is even more robust: the DFC consistently identifies more model exclusive features corresponding to meaningful differences in behavior.
>
> 5. Increased False Positive Rate
> You asked if the increased false positive rate (Figure 3) is generalizable. Yes, this is an expected trade-off. We argue in Section 6.1 that for safety auditing, this high-recall/lower-precision trade-off is acceptable because false positives can be filtered during the validation phase of the "screen-and-verify" workflow.
>
> Thanks in general for the very helpful comments and we remain available to address any remaining comments or concerns!

---

> > ### Comment · Reviewer_H3Af · 2025-11-27
> >
> > Thanks for extensively addressing the points of concern. I will maintain my score.

---

### Official Review · Reviewer_vpNH · 2025-11-09

**Soundness:** 1
**Presentation:** 1
**Contribution:** 1
**Rating:** 0
**Confidence:** 4

**Summary:**

This paper studies model diffing—identifying how two trained models differ by comparing internal representations. The authors adapt Crosscoders to a new variant, Dedicated Feature Crosscoder (DFC), intended to better isolate features unique to one model when architectures differ. Crosscoders learns a shared feature space between different models.

**Strengths:**

Important problem: systematic methods for model diffing across different architectures are valuable.

**Weaknesses:**

Limited novelty w.r.t. Crosscoders, amplified with negligible gains. The method appears to be a modest modification of existing Crosscoders to encourage exclusivity. The paper does not convincingly argue why vanilla Crosscoders are fundamentally unable to isolate exclusive features, nor provide theory or diagnostics showing the failure mode that DFC fixes. Empirical gains (e.g., Fig. 3) are very small, questioning the modification further. Finally, as experiments show, existing Crosscoders work well in this setting where models have different architectures.

Clarity and presentation issues. The exposition is not self-contained. Section 2.1 interleaves prior Crosscoder material with new contributions, obscuring what’s novel. Lines 139–142 are particularly difficult to parse. Key assumptions and the exact DFC objective are underspecified.

Inadequate baselines and analysis. The paper omits a substantial body of prior work that targets the same goal. In particular, it should compare against explanation-based techniques explicitly designed to identify differences between models with disparate architectures (e.g., [jia2022a]). Without these baselines, it’s unclear whether the proposed method offers any advantage over established approaches.

@inproceedings{
jia2022a,
title={A Zest of {LIME}: Towards Architecture-Independent Model Distances},
author={Hengrui Jia and Hongyu Chen and Jonas Guan and Ali Shahin Shamsabadi and Nicolas Papernot},
booktitle={International Conference on Learning Representations},
year={2022},
url={https://openreview.net/forum?id=OUz_9TiTv9j}
}

**Questions:**

What is the exact gain/contribution/novelty of the proposed modification to existing Crosscoders?

Why is Crosscoders the right solution for you to choose? Why not adopt or extend explanation-based techniques explicitly designed for cross-architecture comparisons (e.g., [jia2022a])? Can you provide a principled justification (failure modes, assumptions) and empirical evidence (head-to-head baselines, ablations) supporting this choice?

Can you provide a clear, formal definition of “model-exclusive feature” and a quantitative metric used throughout?

---

> ### Author Response · Authors · 2025-11-24
>
> We thank the reviewer for their detailed comments
>
> We have revised the paper to address your specific concerns regarding novelty, baselines, and clarity, and address some of your concerns below.
>
> 1. Novelty and Magnitude of Gains
> You raised concerns that the DFC is a modest modification with negligible gains. While the architectural change is straightforward, we argue that the functional gains for the specific task of model diffing are significant:
> - Recall (Toy Model): In the data-constrained regime (which reflects real-world training), Figure 3 (left) shows the DFC recovers significantly more ground-truth exclusive concepts than the standard crosscoder.
> - Quality (Real Models): As shown in the new Figure 4 (left), the distribution of exclusivity scores for DFC features is heavily skewed towards 5.0 (highly exclusive), whereas the standard crosscoder is bimodal.
> - Practical Utility: As shown in Table 7 (Appendix C.3), the standard crosscoder failed to identify the granular "Hong Kong," "Taiwan," and "Debt Trap" features that the DFC successfully recovered. We believe this difference in identifying granular, safety-critical narratives constitutes a meaningful practical gain.
>
> 2. Why Standard Crosscoders Fail
> You asked for a diagnostic on why standard crosscoders fail. We have clarified this in Section 3.1. Standard crosscoders optimize a joint reconstruction loss. A feature that reconstructs both models ($R^2$ gain in both terms) is strictly preferred by the loss function over a feature that reconstructs only one, creating a "competition for capacity" where shared features crowd out exclusive ones. The DFC structurally reserves capacity that cannot be claimed by shared features, shielding them from this optimization pressure.
>
> 3. Comparison to Explanation-Based Baselines (Jia et al., 2022)
> Thank you for highlighting this related work. We have added a discussion and citations in Section 5 to distinguish our approaches:
> - Different Objectives: Methods like ModelDiff or Zest of LIME (Jia et al.) produce a scalar distance metric or identify input regions of disagreement. Our goal is to discover the latent variables (features) that constitute that difference (e.g., discovering a specific "Copyright Refusal" direction in activation space).
> - Complementary Tools: We view these methods as complementary. A distance metric tells you that models differ; a crosscoder tells you how (by providing the interpretable feature vector).
>
> 4. Clarity and Formal Definitions
> We have rewritten Section 3 to separate background material better from our contributions. We added a definition of exclusivity in We added Equation 2 to explicitly define the DFC objective function and the partitioning mechanism. We also added a description of key assumptions at the beginning of section 3.1.
>
> We hope these revisions and answers address your concerns and demonstrate the value of the DFC for unsupervised feature discovery, and that you will consider updating your score accordingly.
>
> We remain available to address any remaining questions or concerns.

---

### Author Response · Authors · 2025-11-24
**Comment to all reviewers**

We thank the reviewers for their insightful and constructive feedback. We have taken these comments into account and modified the paper significantly to address concerns regarding baselines, robustness, and clarity. We believe the result is a much stronger paper, for which we are grateful to the reviewers, and ask them to consider updating their scores accordingly

Key Updates in the Revision:

"Screen-and-Verify" Workflow (Section 4.3): We have clarified the intended utility of our method. We explicitly acknowledge that most of the model-exclusive features identified by the DFC do not correspond to meaningful differences in model behavior. Consequently, we propose crosscoder model diffing primarily as a high-recall prescreening tool designed to surface potential areas of divergence for further investigation. The specific features presented in the paper were identified through a multi-step auditing pipeline: filtering for potentially concerning features based on automated explanations, followed by validation that they correspond to real differences in behavior and that they are causally effective using external evaluations and activation steering. We clarify that this is the intended use for our tool at the beginning of Section 4.3.

Discovery of "Unknown Unknowns" (Section 4.3): We highlight a key result to demonstrate this workflow: while the ideological alignment features were hypothesized based on prior work (Rager et al., 2025), the copyright refusal feature in GPT-OSS-20B represents a meaningful behavioral difference that we were unaware of a priori. This discovery highlights the potential of crosscoder model diffing to uncover unknown unknowns in model behavior.

Robustness Analysis (Table 7 & Appendix C.3): We ran the Dedicated Feature Crosscoder (DFC) with multiple random seeds and partition sizes. Result: Broad behavioral features are consistently identified, while granular features are sensitive to partition size, with larger partitions favoring granular discovery. We have added a brief section discussing this in the paper as well as a more detailed breakdown in Appendix C.3.

Extended Baselines (Figure 3): We added the "Designated Shared Feature" (DSF) crosscoder to our toy model evaluation. Result: The DFC outperforms the DSF baseline in recovering model-exclusive concepts.

Random Feature Sampling (Appendix C.5): We added an analysis of randomly sampled model-exclusive features to address cherry-picking concerns.

---

### Meta-Review · Area_Chair_1xZ9 · 2026-01-05

**Summary:**

While the reviewers see some merits in the paper, this submission should not be accepted in its current form due to several fundamental issues, as pointed out by the reviewers, including

- lack of clear motivation on the importance of studying model differences
- presentation and clarity
- limitations of the proposed approach (e.g., high false positive rates)

Overall, this paper requires significant modifications and another round of full review.

**Reviewer Concerns:**

All are outstanding. The authors' rebuttal is insufficient to address the reviewers' concerns.

**Reviewer Scores:**

There were some responses, but none of them indicate changes

---

### Decision · Program_Chairs · 2026-01-26

Reject